# Topological Data Analysis for Multivariate Time Series Data

**DOI:** 10.3390/e25111509

**Published:** 2023-11-01

**Authors:** Anass B. El-Yaagoubi, Moo K. Chung, Hernando Ombao

**Affiliations:** 1Statistics Program, King Abdullah University of Science and Technology, Thuwal 23955, Saudi Arabia; hernando.ombao@kaust.edu.sa; 2Biostatistics and Medical Informatics, University of Wisconsin-Madison, Madison, WI 53706, USA; mkchung@wisc.edu

**Keywords:** topological data analysis, persistence diagram, persistence landscape, multivariate time series analysis, brain dependence networks

## Abstract

Over the last two decades, topological data analysis (TDA) has emerged as a very powerful data analytic approach that can deal with various data modalities of varying complexities. One of the most commonly used tools in TDA is persistent homology (PH), which can extract topological properties from data at various scales. The aim of this article is to introduce TDA concepts to a statistical audience and provide an approach to analyzing multivariate time series data. The application’s focus will be on multivariate brain signals and brain connectivity networks. Finally, this paper concludes with an overview of some open problems and potential application of TDA to modeling directionality in a brain network, as well as the casting of TDA in the context of mixed effect models to capture variations in the topological properties of data collected from multiple subjects.

## 1. Introduction

The field of topology has a rich history spanning more than two centuries, with its origins tracing back to the work of Leonhard Euler on the famous Königsberg bridge problem. Euler tackled this problem in the 18th century, specifically in the year 1736, as he sought to find a solution to the challenge of finding a walk through the city that would cross each of its bridges once and only once [1]. In the centuries that followed, the field of topology was enriched by the contributions of numerous renowned mathematicians, such as Enrico Betti, Camille Jordan, Johann Benedict Listing, Bernhard Riemann, Felix Hausdorff (much later), and many others [2]. By the turn of the 20th century, Henri Poincaré had developed the concepts of homotopy and homology, thus starting the new field of algebraic topology. The field of topology witnessed major advances and theoretical breakthroughs throughout the twentieth century, becoming one of the most important fields of mathematics, but without practical applications.

Despite the major theoretical development throughout the 1900s, the application aspect did not really take off until much later. Indeed, it was only in the early 2000s that topology found its way to the applications arena under the coinage of topological data analysis. Topological data analysis (or TDA) has witnessed many important advances over the last twenty years, it aims to unravel and provide insights about the “shape” of the datum, following the central TDA dogma: datum has a shape, the shape has meaning, and meaning drives value. This is done by analyzing the persistence homology using a persistence diagram or barcode. The reader is referred to [3] for an introduction to the notion of persistence, [4,5] for a survey on persistent homology and barcodes, and [6] for a review of TDA in general.

Several tools have been developed under the TDA framework to analyze many types of data. These tools have been applied to a broad array of scientific fields, including biology [7], finance [8], and brain signals [9,10]. These tools aim to guide the practitioner to understand the geometrical features present in high-dimensional data, which are not always directly accessible using other classical techniques. Even if such features could be observed upon examination using classical graph-theoretical methods, these would not have been found automatically if the topological methods had not first detected them.

In recent years, the field of deep learning has witnessed a growing emphasis on the application of deep neural networks to investigate physiological signals, with a primary focus on brain imaging [11]. Convolutional neural networks (CNNs), pioneered by Yan LeCun and his collaborators [12,13], have been instrumental in driving progress across various scientific domains, including neuroscience [14]. These models have demonstrated their effectiveness in tasks such as the classification and segmentation of neuroimaging data [15,16,17].

Notably, CNNs have been applied to neuroimaging data, including functional MRIs (fMRIs), where the BOLD response is treated as a 3D image at the voxel level [18]. CNNs leverage convolutional filters to extract meaningful features [13]. Additionally, CNN-based approaches have explored regions of interest (ROIs) to reduce spatial complexity and estimate dependence between pairs of ROIs, leading to the analysis of functional connectivity matrices [19,20].

Moreover, the recent advancements in graph neural networks (GNNs) [21,22] have expanded the scope of CNN models to accommodate irregular data structures, such as weighted networks. GNNs harness message-passing techniques, allowing the application of CNN-like models to situations where node neighborhoods are determined by network weights instead of node positions [23,24]. This progress introduces new possibilities for exploring weighted brain networks, providing valuable insights into their topological properties. Nonetheless, the challenges related to limited interpretability and explainability, model selection, the risk of overfitting, sensitivity to node ordering, and the absence of global information present significant barriers to the efficacy of such approaches.

Despite recent efforts to employ GNN-like models for assessing and interpreting topological information in brain networks [25,26], these models exhibit limited descriptive power when contrasted with the insights provided by topological data analysis (TDA) methods. Rather than pursuing one method’s dominance over the other, there is significant potential in merging deep learning and topological data analysis to unravel how topological information is encoded within these networks, as demonstrated in prior work [27].

The primary tools in TDA are Morse and Vietoris–Rips filtrations, which have been extensively used in various applications. For instance, Morse filtrations have been used to study patterns in imaging data, such as in [28,29], or the geometry of random fields in general, as in [30]. However, Vietoris–Rips filtrations (through persistence homology) have been extensively used to study many types of point cloud datasets. This includes time series data and their transformations using various embedding techniques. To better understand persistence homology, a few concepts, such as simplicial complexes and their filtrations, will be introduced.

No matter the chosen approach, whether GNN-based or TDA-based, the analysis of brain signals comes with a diverse set of strengths and limitations. Table 1 offers a concise overview, outlining the distinctive advantages and potential challenges associated with each method.

This paper has the overarching goal of introducing fundamental TDA concepts to scientists working with diverse data types, including biological, physical, and financial signals. It provides a comprehensive perspective on how these concepts can be applied to the analysis of multivariate time series data. The paper begins with a concise introduction to TDA and persistent homology. In Section 2, we delve into crucial TDA background topics, encompassing Morse filtration, persistent homology, and time-delay embeddings for univariate time series. Section 3 is dedicated to the application of TDA to dependence networks in multivariate time series. Section 4 presents a framework for identifying topological group differences via permutation tests, and Section 5 explores some open problems of interest in neuroscience, particularly in studies that investigate variations in brain dependence across disease groups.

## 2. Background Material For TDA

Understanding topological data analysis (TDA) requires familiarization with a few key concepts. In the TDA literature, it is common to find terminology such as data points, distance between points, or whether the following TDA summary is stable or unstable. Such phrases often use vocabulary familiar to the data scientists; however, the meaning may differ greatly, making it difficult for the reader to understand. Therefore, we invite the reader to ponder on the following questions.

Meaning of data: What constitutes data?Meaning of distance: How can we define a meaningful distance (or discrepancy) between data points?Notion of stability: Is this given TDA summary stable? This is addressed through stability theorems.

These questions will be addressed in this section as well as in coming sections. First, we start by investigating the application of TDA to univariate time series data, then we will consider multivariate time series data, such as brain electroencephalogram signals recorded from electrodes on the scalp. Furthermore, we will examine dependence-based distance functions that measure the degree of association between time series components (e.g., between pairs of electrodes). Finally, we demonstrate TDA on real-world applications with dependence networks.

Data encompass a wide range of concepts. In the mind of a geneticist, ‘data’ refer to a distinct entity (e.g., a sequence of nucleotides). However, data may take on different forms for neuroscientists, such as electroencephalogram (EEG), local field potential (LFP), neuronal spike trains, or functional magnetic resonance imaging (fMRI). In general, data can be represented in various forms, such as images, functions, time series, counts, random fields, bandpass-filtered signals, Fourier transforms, localized transforms (e.g., wavelets and SLEX), etc. In other words, every data type may require a different statistical approach. For example, a cloud of points in Euclidean space might require one statistical approach, while a time series of counts might require another.

Similarly, the notion of distance could potentially vary across data modalities. Indeed, the notion of distance is intrinsically linked to the nature of the data. For example, the discrepancy between two DNA sequences may be different from the discrepancy between two EEGs. Distance means some measure of proximity between data points present in some space of reference. For instance, it is meaningless to use a Euclidean distance when dealing with categorical data as opposed to continuous data, e.g., gender data or a count time series vs. a cloud of points in $R^{2}$.

In general, the notion of distance aims to capture some notion of similarity. When the data naturally originates from a meaningful metric space, one can use the inherited distance metric from the space of reference. Otherwise, a more suitable metric should be used to capture the information of interest. For example, if we are interested in studying the brain networks originating from brain imaging techniques, a meaningful distance metric could be based on the notion of dependence. Sometimes, the goal of a study is to examine the extent of synchrony between regions in a brain network and how that synchrony may be disrupted due to an experimental stimulus or a shock (e.g., epileptic seizure). Often, it would be more informative to study the potential cross-interactions between oscillatory activities at different channels and how an increased amplitude in one channel may excite (increase the amplitude) or inhibit (decrease the amplitude) of another channel. The comprehensive and general notion of cross-oscillatory interactions, i.e., spectral dependence, is discussed in [31]. In practice, correlation-based distance measures are commonly used because of their simplicity in computation and interpretation. However, these approaches can only examine linear associations and may not be adequate for brain signals, where frequency-specific cross-interactions and even non-linear interactions may be more informative.

Stability is a tricky concept mainly because it has many facets, especially in the context of TDA. While the origins of TDA have been in mathematics, it is now more often applied and further developed by data scientists and statisticians who may have different motivations and applications in mind. In the mind of a mathematician, the stability of some transformation (*g*) means robustness to small deformations in the input and is usually captured by inequalities, such as
(1)$$\begin{array}{c} d_{F}\left(g(X),g(Y)\right)\leq d_{E}\left(X,Y\right),\;\mathrm{where}\;g:E\to F \end{array}$$
where *Y* is a smooth transformation of *X*; *g* is the transformation for which we want to prove the stability; $d_{E}$ is a distance metric (e.g., Euclidean or Hausdorff) between *X* and *Y*; and $d_{F}$ is a norm, e.g., Frobenius ${\left||.|\right|}_{F}$ if function *g* returns a matrix or the Wasserstein or Bottleneck distance, as defined in Equations (Equation 2) and (3), if *g* return a set of points.
(2)$$\begin{array}{cc} W_{p}\left(A,B\right) & =\underset{\gamma :A↠B}{\inf}\;{\left(\sum_{x\in A}||x-\gamma \left(x\right)|{|}_{\infty }^{p}\right)}^{\frac{1}{p}}, \end{array}$$
(3)$$\begin{array}{cc} W_{\infty }\left(A,B\right) & =\underset{\gamma :A↠B}{\inf}\;\underset{x\in A}{sup}||x-\gamma \left(x\right)|{|}_{\infty }, \end{array}$$
where γ represents a bijection between sets *A* and *B*.

For statisticians and data scientists, stability has a completely different meaning. Indeed, statistical reasoning considers stability based on the robustness of a conclusion, e.g., the result of a statistical test or inference, against perturbations in the original data due to random noise or any departure from the initial model assumptions. When the mathematician considers small/smooth perturbations in the input, the statistician considers adding random noise with small standard deviations or a set of outliers to the data. Furthermore, the statistician might also consider the perturbation in terms of a change in the distribution of the data, e.g., if the distribution of data has a heavier tail (for example, Student’s t-distribution instead of a normal distribution), this might result in the presence of many unexpected outliers.

Both perspectives can lead to the same result in some cases. For instance, if data are sampled from a manifold and small sampling noise is added, this is typically similar to smoothly deforming the manifold and resampling from the new manifold, as seen from Figure 1. Note that, later in Figure 24 we provide an illustration of this abstract notion of sampling time series components from an underlying dependence manifold. However, in this context, adding an outlier to the data may alter the topology of the object completely. In this case, having a stability theorem is of little practical importance since it is too restrictive. Indeed, adding an outlier completely alters the input. Hence, the inequality in Equation (Equation 1) still holds. However, it is not always tight enough to provide meaningful conclusions. Further details regarding the persistence diagram are discussed in [32].

A persistence diagram of a topological space is a multiset of points (birth-death pairs) that represent the various features present in the topology of the dataset. A birth time means that at this specific time or scale a new topological feature appeared in the filtration, and the corresponding death time is the time or scale from which such topological feature is no longer present in the filtration. When dealing with one-dimensional Morse functions, we can only consider zero-dimensional features (number of connected components), but in general, when analyzing an arbitrary set of points, we can consider higher-dimensional features (wholes, voids, etc.) to better capture the shape of the underlying manifold at hand. For example, the persistence diagrams in Figure 2 show how even a little additive noise (with small σ) can perturb the persistence diagram. Adding an outlier to the middle of the circle, however, completely changes the location of the dots (BLUE and ORANGE), which may lead to different conclusions about the underlying unknown process that generated the data. Persistence diagrams help visualize/summarize the topological information contained in the data. A detailed explanation of this notion will follow in the next section.

### 2.1. Persistent Homology of Morse Filtration

The observed time series is often modeled using a mean structure plus a random (and possibly correlated) noise, as seen in Equation (Equation 4) below
(4)$$\begin{array}{c} y(t)=\mu (t)+\epsilon (t) \end{array}$$
where the mean structure μ(t) is supposed to capture the deterministic trend, and the noise ϵ(t) accounts for the stochastic fluctuations around the mean, which captures the autocovariance (or generally, the within dependence) structure. Viewing μ(t) as a Morse function allows us to use the Morse theory to build a sub-level set filtration that captures the topological information contained in μ, specifically the arrangement of its critical values, see Theorem 3.20 in [33], which states the following: The homotopy of the sub-level set only changes when the parameter value passes through a critical point. To summarize, the “homotopy” of a set only considers the critical information about its topology and disregards the effect of continuous deformations; for example, shrinking or twisting the set without tearing.

In general, EEG signals represent the superposition of numerous ongoing brain processes. Therefore, to accurately characterize and estimate brain functional responses to a stimulus or a shock, it is necessary to record many trials of the same repeated stimulus. In this case, a smoothing approach (i.e., averaging many time-locked single-trial EEGs recorded from the same stimulus) is meaningful as it allows random (non-related to the stimulus) brain activity to be canceled out and relevant signals to be enhanced. This new signal is referred to as event-related potential (ERP).

Indeed, applying Morse filtration to ERP data is meaningful as it allows capturing meaningful information regarding the critical values of the ERP signal. On account of the additive nature of the noise ϵ, a first step is needed to smooth the time series (i.e., recover the mean structure μ(t)). After smoothing, the Morse filtration can be built and visualized, as seen in Figure 3.

As parameter *a* increases, the corresponding sequence of pre-images (i.e., $\mu^{-1}\left(\right]-\infty ,a])$) forms a sub-level set filtration. The topology of the pre-image only changes when *a* goes through a critical value with the non-vanishing second derivative or non-singular Hessian matrix. At every critical value, a component is either born (at local minima) or dies (at local maxima) by merging with another component.

The Morse filtration summarizes the topological information contained in the mean of the time series, μ(t), by capturing the information contained in the arrangement of the local extrema. This implies that TDA incorrectly ignores the information contained in the noise structure, e.g., the covariance and dependence structure. Therefore, the practitioner has to verify that the assumptions of Equation (Equation 4) are valid. Otherwise applying the Morse filtration will result in significant information loss. For example, in the time series analysis, such a model could have disastrous consequences, as in the case of autoregressive processes; see Figure 4. In this example, the presence of a high-frequency first-order autoregressive (AR(1)) process does not modify the mean structure significantly, which does not alter the conclusions of the Morse filtration as the smoothing step cancels the noise structure and unravels the true mean structure. The presence of a low-frequency AR(1) noise process can be problematic because this can be incorrectly absorbed into the mean structure μ(t). This could lead the Morse filtration to erroneous conclusions.

### 2.2. Persistent Homology of Vietoris–Rips Filtration

Homology theory, initially developed to differentiate between topological objects using group theory, serves as a fundamental tool in algebraic topology. It enables the analysis of diverse properties associated with these objects, including connected components, voids, and cavities. Figure 5 illustrates various topological objects, such as polygons, spheres, and tori, with each exhibiting unique characteristics with varying Betti numbers. Typically, the data are assumed to be finite samples from a distribution on an underlying topological space. In order to analyze the shapes of the data, we usually build the homology of the data by looking at the generated networks of neighboring data points at varying scales/distances, as seen in Figure 6. We call this sequence of increasing networks the Vietoris–Rips filtration, see [34]. In Figure 6, we demonstrate that with the increasing radius ϵ, the spheres centered around data points expand, creating a network of interconnected points. The objective of this approach is to identify when these geometric patterns emerge (birth) and how long they persist (death time minus birth time) across a broad range of radius values. In this way, TDA provides a solution that obviates the requirement for potentially arbitrary threshold selections.

The Vietoris–Rips filtration is constructed based on the notions of a simplex and simplicial complex, A simplicial complex is a finite collection of sets closed under the subset relation (see illustration in Figure 7 and Figure 8). Simplicial complexes can be considered as higher-dimensional generalizations of graphs. Using simplicial complexes provides a summary of data shapes across all scales through a mathematical object tailored for abstract manipulations.

Simplicial complexes can be as simple as a combination of singleton sets (disconnected nodes), or more complicated, such as a combination of pairs of connected nodes (edges), triplets of triangles (faces), quadruplets of tetrahedrons, any higher-dimensional simplex, as shown in Figure 7), or a combination of different k-simplices in general, as shown in Figure 8. Though the notion of a simplicial complex may seem almost identical to that of networks, there is a major difference. On the one hand, networks and graphs disregard surfaces, volumes, etc., and can be thought of as flexible structures, whereas simplicial complexes take a much richer approach by keeping track of many levels of complexity represented by various k-simplices.

The above definition of simplicial complexes provides a rigorous description of the Vietoris–Rips filtration as an increasing sequence of simplicial complexes. To construct this increasing sequence of simplicial complexes, practitioners use the concept of a cover with an open ball around each node. An important motivation behind this approach is the Nerve theorem [35]. The Nerve theorem, historically proposed by Pavel Alexandrov (sometimes attributed to Karol Borsuk), simplifies continuous topological spaces into abstract combinatorial structures (simplicial complexes) that preserve the underlying topological structure and can be examined by algorithms. The Nerve theorem states that a set and the nerve of the set covering are homotopy equivalent as the resolution of the cover increases, i.e., they have identical topological properties, such as the number of connected components, holes, cavities, etc.

As a result of the Nerve’s theorem, see Figure 9, the topological properties (i.e., Betti numbers) of the simplicial complexes generated from the open cover should emulate those of the underlying manifold as the resolution of the covering increases and, thus, the open cover converges to the original manifold.

In practice, to construct the Vietoris–Rips filtration from a finite set of points, one looks at the increasing finite cover (${⋃}_{i=1}^{n}U_{i}\left(r\right)$, where $U_{i}\left(r\right)$ is a ball centered around the *i*th point/node with radius *r*) of the topological space of interest at a wide range of radius values, as seen in Figure 7. Another example is considered in Figure 10, where the threshold values (where topological features appear or disappear) are denoted by $\epsilon_{i}$. Once the persistent homology is constructed, it needs to be analyzed using some topological summary such as the barcode, the persistence diagram, or the persistence landscape. In Figure 11, we see the representation of the corresponding persistence diagram and the persistence landscape.

The persistence diagram (PD) is constructed based on the times of birth and death of the topological features in the filtration as seen in Figure 11. Thus, for every birth-death pair, a point is represented in the diagram, e.g., ($\epsilon_{1}$, $\epsilon_{2}$) and ($\epsilon_{2}$, $\epsilon_{3}$). The points in the PD are color-coded so that every color represents a specific dimension of the homology (dimension 0 for connected components, dimension 1 for cycles, etc.).

It is difficult to manipulate (e.g., take averages or compute distances) persistence diagrams (e.g., compute Bottleneck distance or Wasserstein distance; see Figure 12). In [36], the authors compare persistence diagrams. In particular, they show how it can be time-consuming to compute the Bottleneck or the Wasserstein distances as it is necessary to find point correspondence. Additionally, defining a “mean” (or center) or “variation” (or measure of spread) of a distribution persistence diagram is not straightforward, especially when the number of points in each diagram varies.

For these reasons, practitioners prefer to analyze a transformation (persistence landscape) of the persistence diagram, which is a simpler object (function), as defined in [37]; see Figure 11. The persistence landscape (PL) can be constructed from the persistence diagram (PD) by drawing an isosceles triangle for every point of a given homology dimension in the PD centered around the birth and death times, as shown in Figure 11. In case there are intersecting lines, the most persistent (highest) function is defined to be the PL (i.e., $\lambda_{1}$); see Figure 13. For more details regarding the properties of the persistence landscape, refer to [37].

Since the PLs are functions of a real variable (scale), the framework enables the computation of group “averages” and to conduct proper statistical inference, such as the construction of confidence regions. Consequently, other statistical properties, such as the strong law of large numbers, and the central limit theorem, can be derived for the PL, as shown in [37].

### 2.3. Time-Delay Embeddings of Univariate Time Series

So far, we described the process of building the persistence homology from a cloud of points in a metric space. However, in order to analyze time series data using the previous method, it is necessary to create some kind of embedding of the univariate time series into a metric space. For example, instead of studying the time series {Y(t),t=1,…,T}, we will study the behavior of the cloud of points $\left\{Z_{s}=\left(Y\left(s\right),Y\left(s-1\right)\right),s=2,\ldots ,T\right\}$, as seen in Figure 15. This particular embedding is known as the time-delay embedding, see [38], and it aims to reconstruct the dynamics of the time series by taking into consideration the information in lagged observations.

In a time-delay embedding, the aim is to reconstruct the phase space (the space that represents all possible states of the system) based on only one observed time series component, borrowing information from the lagged observations to do so. Under the initial assumptions, this is indeed possible since all the components are interdependent through the shared dynamical system. This phase space may contain valuable information regarding the behavior of the time series. For example, the time series might display some chaotic behavior in time; see Figure 14. However, in the phase space, it might show some convergence to some attractor (a region of the phase space toward which the system converges); see [38,39] for more details regarding the application of topological data analysis to time series data. The foundation of this method derives from the framework of dynamical systems. Indeed, Takens’s theorem states conditions under which the attractor of a dynamical system can be reconstructed from a sequence of observations, as can be seen in Figure 14.

Therefore, when practitioners use topological data analysis to analyze the shape of the point cloud embedding of a time series, they are, in fact, assessing the geometry of the attractor of the underlying dynamical system.

This approach is perfectly meaningful if the initial assumptions of Takens’s theorem are valid, which is assuming that there is a corresponding dynamical system that is continuous and invertible, and corresponds to the observed time series. However, very often in time series analysis (such an approach does not make sense, especially when the observed time series is noisy), has a constant mean, and has no apparent relation with any dynamical system and, hence, it is unlikely to observe such patterns. For example in Figure 15, we see that the time embedding does not show any interesting geometrical features when the level of noise is high.

## 3. Topological Methods for Analyzing Multivariate Time Series

Over the last three decades, the analysis of the Human Connectome using various brain imaging techniques, such as functional magnetic resonance imaging (fMRI) and electroencephalography (EEG), has witnessed numerous successes (see [40,41,42,43,44,45,46,47]), discovering the background mechanisms of human cognition and neurological disorders. In this regard, the analysis of the dependence network of a multivariate time series from a topological point of view will have the potential to provide valuable insight.

A multivariate time series might not display any relevant geometrical features in point cloud embedding. However, it might display topological patterns in its dependence network, as seen in Figure 16.

For this example, none of the previous methods can capture the interdependence between different time series components. However, an application of TDA to the time series’ dependence network would directly reveal the cyclic pattern.

Traditionally, due to their stochastic nature, multichannel or multivariate brain signals have often been modeled using their underlying dependence network, i.e., the dependence between brain regions or nodes in a brain network; see [48,49]. The brain network is often constructed, starting from some connectivity measure derived from the observed brain signals. There are many possible characterizations of dependence and, hence, many possible connectivity matrices. These include cross-correlations, coherence, partial coherence, and partial directed coherence. For a comprehensive treatment of spectral dependence, including potential non-linear cross-oscillatory interactions, we refer the reader to [31].

Due to the difficulty of analyzing and visualizing weighted networks, practitioners apply some thresholding techniques to create a binary network (from the weighted brain network) where the edge exists when the continuous connectivity value exceeds the pre-specified threshold; see [9,50,51,52]. This thresholding is often arbitrarily selected. A major problem associated with this common approach is the lack of principled criteria for choosing the appropriate threshold. Moreover, the binary network derived from a single threshold might not fully characterize the dependence structure. Obviously, the thresholding approach induces a bias and a significant loss of information, leading to a simplified network.

Consequently, applying topological data analysis directly to the original data (i.e., consider a filtration of connectivity graphs) appears to be an appealing alternative to the arbitrary thresholding of weighted networks. First, the TDA approach detects all potential topological patterns present in the connectivity network. Second, by considering all possible thresholding values, TDA avoids the arbitrary thresholding problem. Refer to [9] for more details regarding this problem.

To formalize these ideas, let $X_{i}\left(t\right)$ be a time series of brain activity at location i∈V and time tt∈{1,…T}, where V={1,⋯,P} is the set of all *P* sensor locations (in EEGs) or brain regions (in fMRI). Therefore, considering a set of *P* brain channels (e.g., electrodes/tetrodes) indexed by *V*, the object X=(V,D) is a metric space, where $D_{ij}$ is the dependence-based distance between channel *i* and channel *j* (i.e., between $X_{i}\left(t\right)$ and $X_{j}\left(t\right)$). We build the Vietoris–Rips filtration by connecting nodes of X that have a distance of less or equal to a given ϵ, which results in the following filtration:(5)$$\begin{array}{c} X_{\epsilon_{1}}\subset X_{\epsilon_{2}}\subset \cdots \subset X_{\epsilon_{n}}, \end{array}$$
where $\epsilon_{1}<\epsilon_{2}<\cdots <\epsilon_{n-1}<\epsilon_{n}$ are the distance thresholds. Nodes within a given distance $\epsilon_{i}$ are connected to form different simplicial complexes, $X_{\epsilon_{1}}$ is the first simplicial complex (single nodes), and $X_{\epsilon_{n}}$ is the last simplicial complex (all nodes connected, i.e., a clique of size *n*). In general, Xϵ, for a given ϵ, represents the simplicial complex thresholded at distance ϵ. However, Xϵ only changes for a finite number of distance values, specifically those present in the distance function, i.e., there are at most n=P(P−1)/2 simplicial complexes in the filtration (this is the number of simplicial complexes in the filtration; of course, the number of possible simplicial complexes must be much higher: $2^{\frac{P(P-1)}{2}}$). For a detailed review on how to build the Vietoris–Rips filtration based on a metric space, refer to [34].

Given a topological object X with a filtration, as defined in Equation (Equation 5), the corresponding homology analyzes the object X by examining its *k*-dimensional holes through the *k*-th homology groups $H_{k}\left(X\right)$. The zero-dimensional holes represent the connected components or the clustering information; the one-dimensional holes represent loops, and the two-dimensional holes represent voids, etc. The rank $\beta_{k}$ of $H_{k}\left(X\right)$ is known as the *k*-th Betti number; see the illustration in Figure 5. Refer to [53,54] for more rigorous definitions of these topological objects.

### 3.1. Examples of Time Series Models

In order to show how TDA can be applied to analyze the dependence pattern of a real multivariate time series, we illustrate via simulations how topological patterns, such as cycles and holes, can arise in multivariate time series. Based on the idea, developed in [31,55], the simulated brain signals will be constructed as mixtures of latent frequency-specific oscillatory sources, i.e., a mixture of frequency-specific neural oscillations. Each of these oscillatory random processes can be modeled by a second-order autoregressive (AR(2)) process and, thus, the simulated EEG will be a mixture of these AR(2) processes. A latent oscillatory process with a spectral peak at the alpha band (8–12 Hz) can be characterized by an AR(2) process of the following form:(6)$$\begin{array}{cc} Z^{\alpha }\left(t\right) & =\phi_{1}^{\alpha }Z^{\alpha }\left(t-1\right)+\phi_{2}^{\alpha }Z^{\alpha }\left(t-2\right)+W^{\alpha }\left(t\right) \end{array}$$
where $W^{\alpha }\left(t\right)$ is white noise with $E\;W^{\alpha }\left(t\right)=0$ and $Var\;W^{\alpha }\left(t\right)=\sigma_{\alpha }^{2}$; and the AR(2) coefficients $\phi_{1}^{\alpha }$ and $\phi_{2}^{\alpha }$ are derived as follows. Note that Equation (Equation 6) can be rewritten as $W^{\alpha }\left(t\right)=\left(1-\phi_{1}^{\alpha }B^{1}-\phi_{2}^{\alpha }B^{2}\right)Z^{\alpha }\left(t\right)$, where the back-shift operator $B^{k}Z^{\alpha }\left(t\right)=Z^{\alpha }\left(t-k\right)$ for k=1,2. The AR(2) characteristic polynomial function is:(7)$$\begin{array}{c} \Phi \left(r\right)=1-\phi_{1}^{\alpha }r^{1}-\phi_{2}^{\alpha }r^{2}. \end{array}$$
Consider the case where the roots of the Φ(r), denoted by $r_{1}$ and $r_{2}$, are both (non-real) complex-valued and, hence, can be expressed as $r_{1}=M\exp\left(i2\pi \psi \right)$ and $r_{2}=M\exp\left(-i2\pi \psi \right)$, where the phase ψ∈(0,0.5) and the magnitude M>1 satisfy causality [56]. For this latent process $Z^{\alpha }\left(t\right)$, suppose that the sampling rate is denoted by SR and the peak frequency is $f_{\alpha }\in$(8–12 Hz). Then the roots of the AR(2) latent process are $r_{1}^{\alpha }=M_{\alpha }\exp\left(i2\pi \psi_{\alpha }\right)$ and $r_{2}^{\alpha }=M_{\alpha }\exp\left(-i2\pi \psi_{\alpha }\right)$ where the phase $\psi_{\alpha }=f_{\alpha }/SR$. In practice, we can choose $\psi_{\alpha }=10/100$ for a given SR=100 Hz and the root magnitude is $M_{\alpha }$ or a number greater than 1 but “close” to 1, so that the spectrum of $Z^{\alpha }\left(t\right)$ is mostly concentrated on the alpha band (8–12 Hz). The corresponding AR(2) coefficients are $\phi_{1}^{\alpha }=\frac{2}{M_{\alpha }}\cos\left(2\pi \psi_{\alpha }\right)$ and $\phi_{2}^{\alpha }=-\frac{1}{M_{\alpha }^{2}}$. An example of such a stationary AR(2) process can be visualized in Figure 17.

**Examples.** The goal here is to illustrate the previous idea by considering multivariate stationary time series data with a given cyclic dependence network (cyclic frequency-specific communities), see Figure 18 and Figure 19. However, the advantage of applying TDA to the weighted network, as explained above, is to detect the presence of such topological features. The simulated time series can be visualized in Figure 20 and Figure 21.

It is very common in brain signals to exhibit communities or cyclic structures, as seen in [46] or in simulations in Figure 22. There are various reasons that could explain the presence of such patterns in brain networks. In particular, the brain network could be organized in such a way as to increase the efficiency of information transfer or minimize the energy consumption. Also, the brain connectivity network could be altered due to a neurological disease, e.g., among patients with Alzheimer’s disease, brain volumes can shrink, and there is severe demyelination that could result in the weakening of structural connections, which could lead to the creation of cycles/holes or voids in the brain’s functional network. In general, we can imagine the following scenarios:Groups of neurons firing together (presence of clusters);Groups of neurons sharing some latent processes (potential cycles).

To generate the previous multivariate time series (in Figure 20 and Figure 21) with both dependence patterns, we use the following approach.

**Example 1:** The goal of this example is to show how to generate a time series with the dependence pattern presented in Figure 18. Let $Y\left(t\right)={\left[Y_{1}\left(t\right),\cdots ,Y_{9}\left(t\right)\right]}^{\prime }$ be the observed time series, $Z\left(t\right)={\left[Z_{1}\left(t\right),\cdots ,Z_{8}\left(t\right)\right]}^{\prime }$ be the latent AR(2) processes (as in Figure 17), and $\epsilon \left(t\right)={\left[\epsilon_{1}\left(t\right),\cdots ,\epsilon_{9}\left(t\right)\right]}^{\prime }$ be the iid Gaussian innovations. Then we can simulate Y(t) as follows:(8)$$\begin{array}{cc} A & =\frac{1}{2}\left(\begin{array}{cccccccc} 1 & 1 & 0 & 0 & 0 & 0 & 0 & 0\\ 0 & 1 & 1 & 0 & 0 & 0 & 0 & 0\\ 0 & 0 & 1 & 1 & 0 & 0 & 0 & 0\\ 1 & 0 & 0 & 1 & 0 & 0 & 0 & 0\\ 0 & 0 & 0 & 1 & 1 & 0 & 0 & 0\\ 0 & 0 & 0 & 0 & 1 & 1 & 0 & 0\\ 0 & 0 & 0 & 0 & 0 & 1 & 1 & 0\\ 0 & 0 & 0 & 0 & 0 & 0 & 1 & 1\\ 0 & 0 & 0 & 0 & 1 & 0 & 0 & 1 \end{array}\right), \end{array}$$(9)$$\begin{array}{cc} Y(t) & =AZ(t)+\epsilon (t). \end{array}$$
where ϵ(t) are iid Gaussian white noises with covariance $\sigma_{\epsilon }=I_{9}$.

**Example 2:** The goal of this example is to show how to generate a time series with the dependence pattern presented in Figure 19. Let $Y\left(t\right)={\left[Y_{1}\left(t\right),\cdots ,Y_{9}\left(t\right)\right]}^{\prime }$ be the observed time series, and $Z\left(t\right)={\left[Z_{1}\left(t\right),\cdots ,Z_{5}\left(t\right)\right]}^{\prime }$ be another set of independent latent AR(2) processes (as in Figure 17). Then using the following matrix, we can generate the second dependence structure in Y(t).
(10)$$\begin{array}{cc} A & =\frac{1}{2}\left(\begin{array}{ccccc} 1 & 1 & 0 & 0 & 0\\ 0 & 1 & 1 & 0 & 0\\ 0 & 0 & 1 & 1 & 0\\ 1 & 0 & 0 & 1 & 0\\ 0 & 0 & 0 & 1 & 1\\ 0 & 0 & 0 & 0 & 2\\ 0 & 0 & 0 & 0 & 2\\ 0 & 0 & 0 & 0 & 2\\ 0 & 0 & 0 & 0 & 2 \end{array}\right), \end{array}$$
(11)$$\begin{array}{cc} Y(t) & =AZ(t)+\epsilon (t) \end{array}$$
where ϵ(t) are iid Gaussian white noises with covariance $\sigma_{\epsilon }=I_{9}$. In order to build the Vietoris–Rips filtration, a dependence-based distance function needs to be defined between the various components of the time series. For instance, a decreasing function of any relevant dependence measure could be useful. Therefore, based on the dependence network, a distance matrix can be used to build the persistence homology. First, we define the Fourier coefficients and the smoothed periodogram as follows: (12)$$\begin{array}{cc} d(\omega_{k}) & =\frac{1}{\sqrt{T}}\sum_{t=1}^{T}X\left(t\right)\exp\left(-i\omega_{k}t\right) \end{array}$$(13)$$\begin{array}{cc} \hat{f}\left(\omega_{k}\right) & =\sum_{\omega }k_{h}\left(\omega -\omega_{k}\right)d\left(\omega \right){d(\omega )}^{*} \end{array}$$
where $k_{h}\left(\omega -\omega_{k}\right)$ is a smoothing kernel centered around $\omega_{k}$ and *h* is the bandwidth parameter. Second, we define the dependence-based distance function to be a decreasing function (e.g., G(x)=1−x) of the coherence: (14)$$\begin{array}{cc} C\left(X_{i}\left(.\right),X_{j}\left(.\right),\omega \right) & =\frac{|f_{i,j}{\left(\omega \right)|}^{2}}{f_{i,i}\left(\omega \right)f_{j,j}\left(\omega \right)}\in \left[0,1\right]. \end{array}$$(15)$$\begin{array}{cc} D\left(X_{i}\left(.\right),X_{j}\left(.\right),\omega \right) & =G\left(C(X_{i}\left(.\right),X_{j}\left(.\right),\omega )\right). \end{array}$$
Coherence at a pre-specified frequency band involves the squared maximal cross-correlation (across phase shifts) between a pair of filtered signals (where power is concentrated at the specific band); see [57]. Another way to estimate coherence is via the maximal cross-correlation-squared of the bandpass filtered signals; see [31].

**Example 3:** The goal of this example is to provide a simple model that can explain/distinguish the observed cyclic structure from the random one in some datasets; see Figure 22. Assume that we observe six time series components that share common cyclic latent independent process copies $Z_{i}\left(t\right)$ for group 1: $$\begin{array}{cc} y_{1}\left(t\right) & =Z_{6}\left(t\right)+Z_{1}\left(t\right)+c\epsilon_{1}\left(t\right)\\ y_{2}\left(t\right) & =Z_{1}\left(t\right)+Z_{2}\left(t\right)+c\epsilon_{2}\left(t\right)\\ y_{3}\left(t\right) & =Z_{2}\left(t\right)+Z_{3}\left(t\right)+c\epsilon_{3}\left(t\right)\\ y_{4}\left(t\right) & =Z_{3}\left(t\right)+Z_{4}\left(t\right)+c\epsilon_{4}\left(t\right)\\ y_{5}\left(t\right) & =Z_{4}\left(t\right)+Z_{5}\left(t\right)+c\epsilon_{5}\left(t\right)\\ y_{6}\left(t\right) & =Z_{5}\left(t\right)+Z_{6}\left(t\right)+c\epsilon_{6}\left(t\right) \end{array}$$
Assume that we observe five time series components that share random common latent independent process copies $Z_{i}\left(t\right)$ for group 2:$$\begin{array}{cc} y_{1}\left(t\right) & =Z_{6}\left(t\right)+Z_{3}\left(t\right)+Z_{4}\left(t\right)+c\epsilon_{1}\left(t\right)\\ y_{2}\left(t\right) & =Z_{6}\left(t\right)+Z_{2}\left(t\right)+c\epsilon_{2}\left(t\right)\\ y_{3}\left(t\right) & =Z_{1}\left(t\right)+c\epsilon_{3}\left(t\right)\\ y_{4}\left(t\right) & =Z_{1}\left(t\right)+Z_{2}\left(t\right)+Z_{3}\left(t\right)+c\epsilon_{4}\left(t\right)\\ y_{5}\left(t\right) & =Z_{4}\left(t\right)+Z_{5}\left(t\right)+c\epsilon_{5}\left(t\right)\\ y_{6}\left(t\right) & =Z_{5}\left(t\right)+c\epsilon_{6}\left(t\right) \end{array}$$
when the parameter c=0 (vanishing noise), the coherence between the observed time series is maximal: Coh$\left(y_{i},y_{j},\omega \right)=1,\forall \omega \in \Omega_{\alpha }$, and as the parameter *c* increases, the coherence between the observed time series drops to zero when c=∞. In this way, the signal-to-noise ratio (SNR=$\frac{Var(Signal)}{Var(Noise)}$) controls the coherence between the components and, therefore, controls the distance between the components. This simple example aims to demonstrate a potential explanation of the mechanism behind the appearance of topological features in the brain dependence network.

After generating the time series data from both models as described previously, we estimate the coherence matrix based on the smoothed periodogram (rectangular window) for the 100–200 Hz frequency band. Based on this, we construct the distance matrix and apply TDA to obtain the PD in Figure 23. We can clearly see that the subplot on the left displays one-dimensional features (orange dot far from the diagonal), whereas the subplot on the right does not.

To summarize, the previous examples intended to convey the underlying topological pattern of the dependence space. We can try to visualize this idea using Figure 24.

### 3.2. TDA vs. Graph-Theoretical Modeling of Brain Connectivity

The human brain is organized both structurally and functionally as well into complex networks. This complex structure allows both the segregation and integration of information processing. The classical approach to analyzing brain functional connectivity consists of using tools from the science of networks, which often involves the use of graph-theoretical summaries, such as modularity, efficiency, and betweenness; see [58]. Graph-theoretical models witnessed an important success in modeling complex brain networks, as described in [59,60]. The above-mentioned graph’s theoretical summaries aim to characterize the topological properties of the network being studied and, hence, can distinguish between small-world, scale-free, and random networks. Indeed, such graph-theoretical models display important features that are of particular interest in the study of brain activity. For example, random networks (a network where the edges are selected randomly) usually have a low clustering coefficient (a low measure of the degree to which nodes in a graph tend to cluster together) and a low characteristic path length (low average distance between pairs of nodes), on the other hand, regular networks have a high clustering coefficient but a high characteristic path length. However, small-world models have high clustering (higher than random graphs) and a low characteristic path (roughly the logarithm of the size of the graph). Furthermore, scale-free networks can have even smaller characteristic path lengths and potentially smaller clustering coefficients than those of small-world networks. Such topological properties may have a direct impact on brain activity, such as the robustness to brain injury or efficiency of information transfer between brain regions that are far apart (variable cost of brain integration). See [61] for an overview of small-world networks and their potential applications and properties and [62] for an overview of the emergence of scale-free networks in random networks. Such models and graph summaries have been extensively used to study the impact of diseases on the topology of brain connectivity; see [63,64].

The goal of such an approach is to characterize the topological properties of the brain’s network. Although such summaries provide interesting and valuable insights into the topology of brain networks, they nevertheless suffer some limitations. Indeed, such summaries cannot capture all topological information contained in the network, such as the presence of holes and voids; see Figure 26. Furthermore, such summaries cannot be applied directly to a weighted connectivity network. Very often, a thresholding step is necessary, which can be a serious limitation because it can result in an important loss of information if the threshold is not selected properly, as seen in Figure 25. In this regard, applying TDA to brain connectivity could provide complementary information on the topology of the brain’s functional network since TDA considers all potential threshold values.

There are many advantages to using the persistence homology techniques. The topological data analysis is designed to study the topological features (geometry and spatial organization) of networks. Classical approaches describe the topological properties of the network. However, it remains difficult to detect/assess the topological patterns present in general, as seen in Figure 26. A graph-theoretical algorithm will count as three different cycles, the cycles around the same hole (green, blue, and red). However, TDA can detect exactly only one large hole/topological feature because it uses the concept of a simplicial complex. Furthermore, an algorithm that clusters the network nodes (modularity analysis) needs a parameter choice, whereas the TDA techniques provide overall answers regarding the network topology without parameter tuning.

## 4. EEG Analysis and Permutation Testing

The purpose of this section is to compare the differences in the topological features of the brain connectivity networks of young individuals with attention deficit hyperactivity disorder (ADHD) and control groups; see [65], specifically, the impact of ADHD on the connected components (0-dimensional homology) and the network cyclic information (1-dimensional homology).

The participants in this study were 61 children with ADHD and 60 controls aged between 7 and 12 years old. The ADHD children were on the drug Ritalin for up to 6 months from the start of the study. None of the children in the control group had a history of psychiatric disorders, epilepsy, or any report of high-risk behaviors. EEG signals were recorded based on 10–20 standard by 19 channels at a sampling frequency of 128 Hz, see Figure 27.

Since visual attention deficits are common characteristics of children with ADHD, the EEG recordings in this study were conducted while the participants engaged in a visual attention task. During this task, the children were presented with a series of cartoon characters and instructed to count them. To ensure a continuous stimulus during the EEG recording, each image was displayed immediately after the child’s response without interruptions. Consequently, the duration of the EEG recording for this cognitive visual task varied, depending on each child’s response speed. We selected 51 subjects with ADHD and retained 53 healthy control subjects for analysis after preprocessing the data using the PREP pipeline. The preprocessing steps included removing electrical line effects, addressing artifacts caused by eye movements, blinks, or muscular activity, identifying and rectifying issues with low-quality channels, filtering out irrelevant signal components, and ultimately enhancing topographical localization by re-referencing the signal.

We compute the average coherence matrix for different frequency bands in Hertz; delta = (0.5–4 Hz), theta = (4, 8 Hz), alpha = (8–12 Hz), beta = (12–30 Hz), and gamma = (30–50 Hz). The distance function was computed to be (D=1−C), as defined in Equation (15). For every frequency band, we build the persistence landscapes for both ADHD and healthy control groups; see Figure 28 and Figure 29.

In Figure 28 and Figure 29, we observe group-level variations in the persistence landscapes. In zero-dimensional homology, these differences are noticeable at the delta, theta, and alpha frequency bands. Similarly, one-dimensional homology exhibits group disparities across all frequency bands, except the gamma band. However, in the case of two-dimensional homology, the differences between the groups appear to be of significantly smaller magnitude, approximately one order of magnitude lower than the previous ones.

Now, if we aim to assess variations in the brain connectivity network topology between the two groups (ADHD and control), we can formulate the null hypothesis $H_{0}$ as follows: “There is no difference in the brain connectivity network between the ADHD and control groups”. To test this hypothesis, we can employ a permutation test based on the norm of the discrepancy persistence landscapes at a specific homology dimension and for a given frequency band Ω”.
(16)$$\begin{array}{c} T_{k}^{\Omega }=\int_{\Omega }||{\bar{\lambda }}_{k}^{\left(1\right)}\left(\omega \right)-{\bar{\lambda }}_{k}^{\left(2\right)}\left(\omega \right)|{|}_{2}d\omega \end{array}$$
To decide whether to reject $H_{0}$, we need to compare the observed test statistic with a threshold obtained from the reference distribution of the test statistic under $H_{0}$. We use a permutation approach to derive this empirical distribution under the null hypothesis, as was conducted in [29,37,66]. A formal framework for testing between two groups in the topological data analysis is presented in [67], with an extension to three groups in [68]. Practical examples of nonparametric permutation tests at an acceptable level can be found in [69]. Refer to [66] for more examples regarding permutation and randomization tests in functional brain imaging and connectivity. Therefore, following the permutation approach, we propose the following procedure:Compute the sample test statistic from the original PLs: $\lambda_{1}^{\left(1\right)},\cdots ,\lambda_{n_{1}}^{\left(1\right)}$ and $\lambda_{1}^{\left(2\right)},\cdots ,\lambda_{n_{2}}^{\left(2\right)}$.Permute the ADHD and healthy control group labels to obtain $\lambda_{1}^{\left(1*\right)},\cdots ,\lambda_{n_{1}}^{\left(1*\right)}$ and $\lambda_{1}^{\left(2*\right)},\cdots ,\lambda_{n_{2}}^{\left(2*\right)}$.Compute the sample discrepancy from the permuted PLs:
$$T^{\Omega *}=\int_{\Omega }||{\bar{\lambda }}_{k}^{\left(1*\right)}\left(\omega \right)-{\bar{\lambda }}_{k}^{\left(2*\right)}\left(\omega \right)|{|}_{2}d\omega$$.Repeat steps 2 to 3, *B* times.Compute the threshold τ as the (1−α)-quantile of the empirical distribution of test statistic ${\hat{F}}_{T}$.

After applying the above to our data, we obtain the following reference distribution for the zero- and one-dimensional homology persistence landscapes.

Despite the differences between the PLs of the two populations in zero- and one-dimensional homology groups, only the differences in the alpha- and beta-frequency bands seem to be significant for the one-dimensional homology group, as shown in Figure 30 and Figure 31.

## 5. Open Problems

As the field of topological data analysis keeps advancing and developing, new and challenging problems continue to emerge. We briefly discuss three open problems that may be of interest to readers with an interest in topological data analysis applied to brain networks.

The study of brain signals shows that brain dependence networks may display between-group discrepancy as well as within-group variability. Historically, linear mixed-effect models (LMEMs) have been proposed to analyze data with fixed effects (average persistence landscape) and random effects (variance of the persistence landscape), e.g., y=Xβ+Uz+ϵ). Is it possible to develop such a model, which can be applied to detect group-level differences (fixed effect, i.e., differences in β) in the topological structure of the network via the estimated persistence landscapes, as well as within-group variations of the topological structure (random effects, i.e., differences in *z*).

Brain dependence networks can be constructed based on various dependence measures. When correlation or coherence are used to measure dependence between brain channels, the resulting dependence network is a non-oriented graph. In contrast, when more complex models of dependence (e.g., the flow of information) are used to model the dependencies between brain channels, such as partial directed coherence, the resulting network is an oriented one. This results in a non-symmetric distance function, which is a problem for the application of TDA. One potential approach to extend the use of TDA to oriented networks is to use the matrix decomposition $A=A^{s}+A^{n}$, where $A^{s}=\frac{1}{2}\left(A+A^{{}^{\prime }}\right)$ and $A^{n}=\frac{1}{2}\left(A-A^{{}^{\prime }}\right)$.

The classical application of TDA results in a global analysis of the network. Therefore, it is impossible to state where the topological features are located in the network. Therefore, it is natural to wonder if TDA can be applied locally to “local sub-networks”. Can we think of TDA in hierarchical terms? Similarly, sometimes a transient change in connectivity can be observed in brain networks (e.g., localized behavior in time that leads to task-specific functional brain connectivity). How can we use TDA to study such evolutionary or transient events?

## 6. Conclusions

Historically, brain network analysis relied on graph-theoretical measures, such as clustering coefficients, betweenness centrality, and the average shortest path length to study the topology. Although such an approach revealed some interesting facts about the brain in the past, it does not provide us with the full picture of the network’s geometry. In contrast, TDA has begun to be used to analyze brain network topological data from a persistent homology perspective. This enables a summary of all the scales without having to use arbitrary thresholds.

The purpose of this paper was to provide a pedagogical introduction to topological data analysis within a multivariate spectral analysis of time series data. This approach has the advantage of combining the power of TDA with spectral analysis, which will allow practitioners to characterize the commonalities and differences in the shapes of brain connectivity networks across different groups for frequency-specific neural oscillations.

We demonstrated the advantages of using the Vietoris–Rips filtration over the Morse one. We presented a pedagogical review of persistent homology using the Vietoris–Rips filtration over a cloud of points. We discussed how time-delay embedding could be pertinent and showed its limits when the initial assumptions of Takens’s theorem are not satisfied. Finally, we recommended applying TDA to the connectivity network, as this could capture the rich information contained in the dependence structures of brain signals, as shown in the data application.

Indeed, the application of TDA to the connectivity networks of ADHD vs. healthy control individuals shows significant discrepancy between their respective PLs at the alpha- and beta-frequency bands for the one-homology group. This suggests that the ADHD condition affects the cyclic structure of the brain connectivity network more than the connected components.

## Figures and Tables

**Figure 1 entropy-25-01509-f001:**
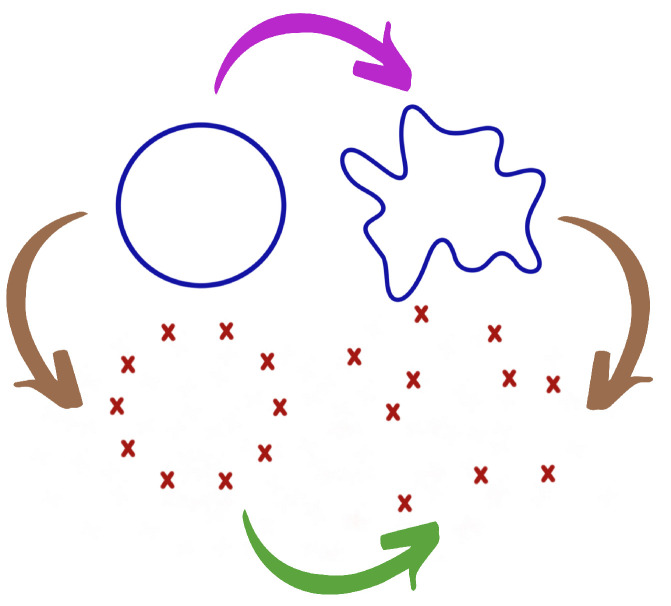
Perturbation of the data from the perspectives of statistics and topology. The original topological structure is in the upper-left corner (circle). The observed cloud of points in the bottom-right corner. The mathematician first considers a perturbation of the manifold (PURPLE arrow) and then a sampling step (BROWN right arrow), whereas the statistician first considers a sampling step (BROWN left arrow) and then the addition of noise (GREEN arrow).

**Figure 2 entropy-25-01509-f002:**
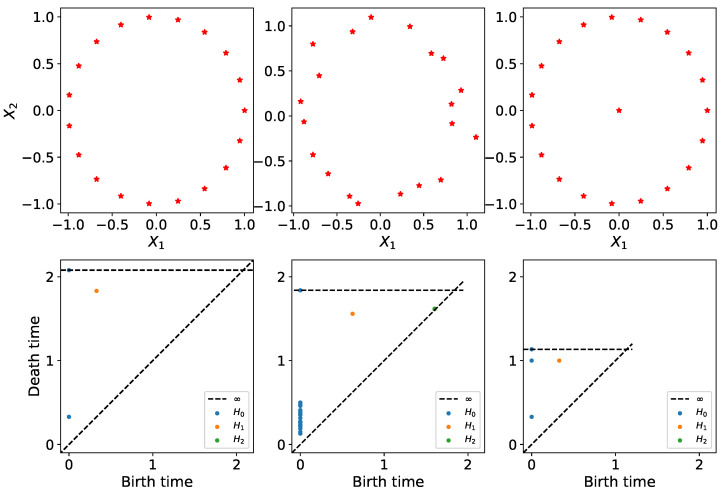
Persistence diagrams of clouds of points. First row: point data cloud, which includes the original data sampled from a circle (**LEFT**); original data plus noise (**MIDDLE**); the original data plus an outlier in the center (**RIGHT**). Second row: corresponding persistence diagram for each cloud of point. This figure demonstrates the sensitivity of the persistence diagram in the presence of outliers. This suggests the need for pre-processing (or some transformation) of the data prior to the application of TDA.

**Figure 3 entropy-25-01509-f003:**
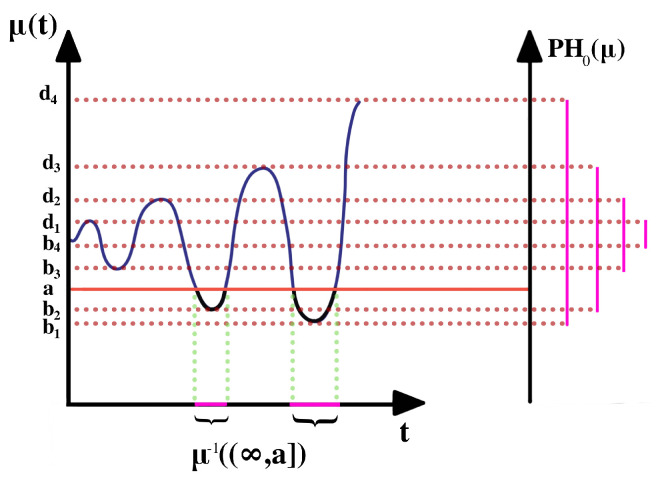
Sub-level set technique for a one-dimensional Morse function. On the left is the function μ(t), on the right is the barcode summary of the zero-dimensional persistence homology.

**Figure 4 entropy-25-01509-f004:**
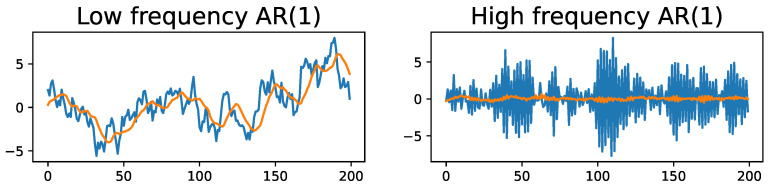
(**LEFT**) Time series with zero-mean plus a low-frequency AR(1) process (ρ=+0.95); original time series in blue and smoothed time series in orange. (**RIGHT**) time series with zero-mean plus a high-frequency AR(1) process (ρ=−0.95); original time series in blue and smoothed time series in orange.

**Figure 5 entropy-25-01509-f005:**
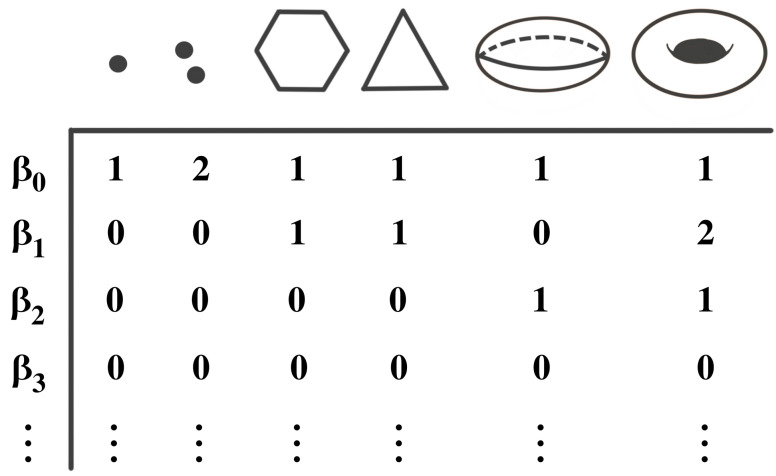
Examples of topological objects with the corresponding Betti numbers. The zero-Betti number ($\beta_{0}$) counts the number of components, the one-Betti number ($\beta_{1}$) counts the number of cycles or wholes, and the two-Betti number ($\beta_{2}$) counts the number of voids, etc.

**Figure 6 entropy-25-01509-f006:**
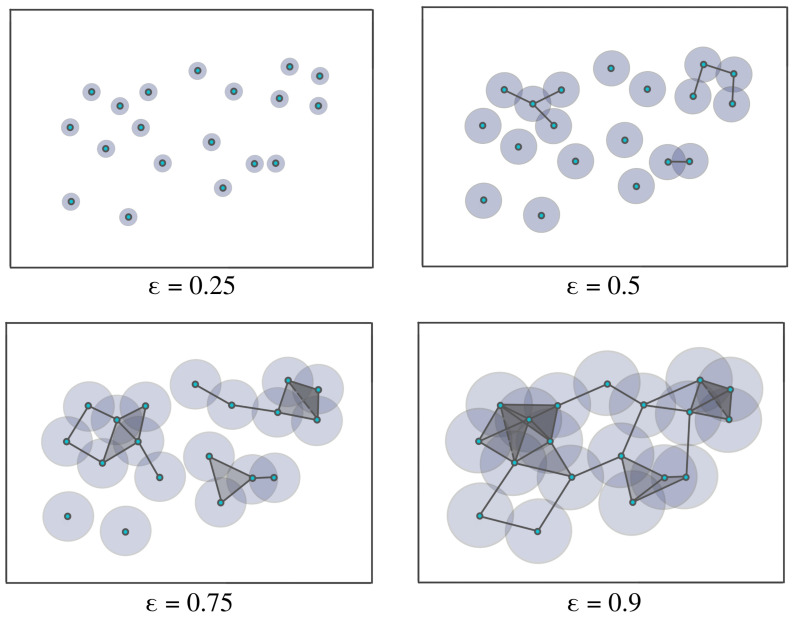
Example of a Vietoris–Rips filtration. As the radius ϵ increases, the balls centered around the data points with radius ϵ start intersecting, leading to the appearance of more features in the increasing sequence of simplicial complexes.

**Figure 7 entropy-25-01509-f007:**
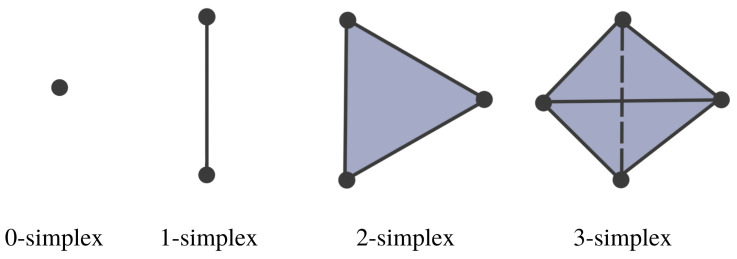
Simplices with different dimensions.

**Figure 8 entropy-25-01509-f008:**
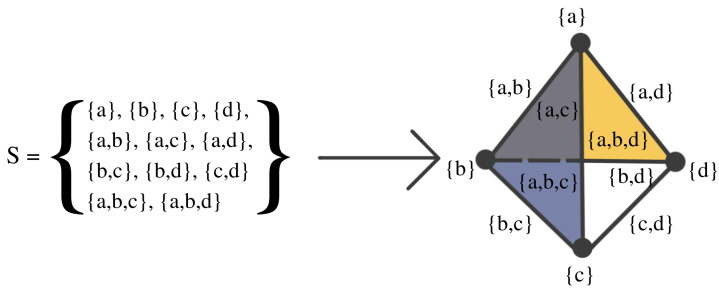
Example of a simplicial complex with four nodes, six edges, and two faces. If *S* is a simplicial complex, then every face of a simplex in *S* must also be in *S*.

**Figure 9 entropy-25-01509-f009:**
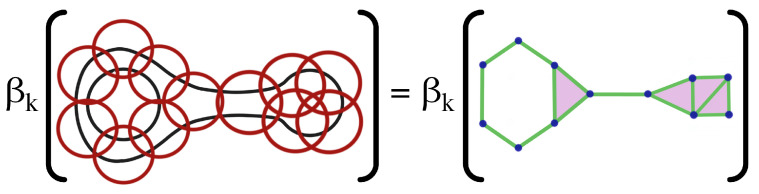
The open cover (**LEFT**) and the corresponding nerve (**RIGHT**) have identical Betti numbers, denoted by $\beta_{k}$ (i.e., number of connected components, holes, voids, etc.). As the resolution of the cover increases the topological structure of the resulting nerve resembles that of the original space.

**Figure 10 entropy-25-01509-f010:**
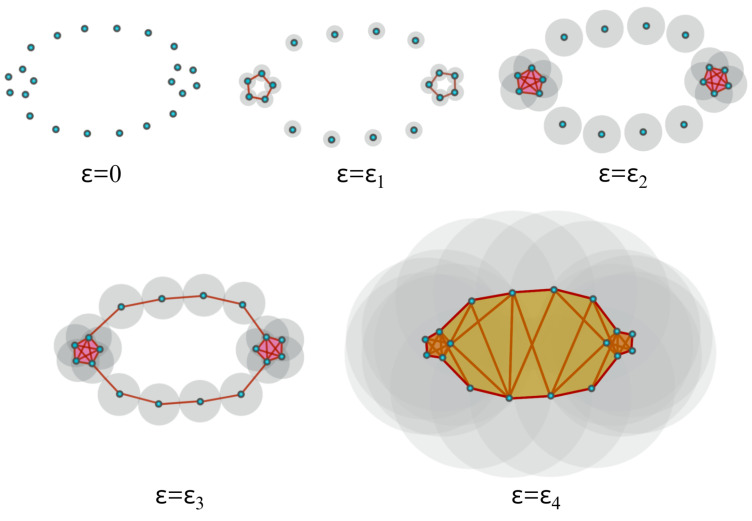
When ϵ=0, all points are disconnected, but as ϵ grows, the open cover becomes larger. When $\epsilon =\epsilon_{1}$, some of the balls start to intersect and, thus, an edge is created (or more generally, a higher-dimensional simplex, if more than two balls intersect) between the pair of points, which results in the creation of a cycle (one-dimensional hole). When $\epsilon =\epsilon_{2}$, more edges (ten 1-simplices) are added, as well as a tetrahedron (four 2-simplices and one 3-simplex), which results in the creation of a new cycle and the destruction of the first cycle. Finally, when $\epsilon =\epsilon_{3}$, more simplices are added and the second cycle disappears.

**Figure 11 entropy-25-01509-f011:**
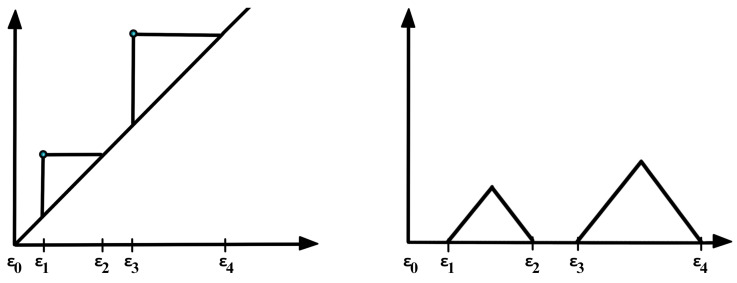
Corresponding persistence diagram (**LEFT**) and persistence landscape (**RIGHT**) from the cloud of points defined in Figure 10.

**Figure 12 entropy-25-01509-f012:**
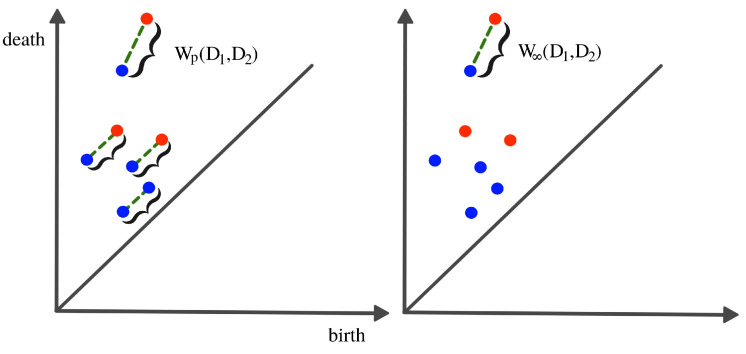
Wasserstein distance (**LEFT**) vs. bottleneck distance (**RIGHT**). To compute any of the two distances, the optimal point correspondence needs to be found, which might become computationally infeasible as the number of mappings increases in the order of $O(n_{E}^{n_{F}})$.

**Figure 13 entropy-25-01509-f013:**
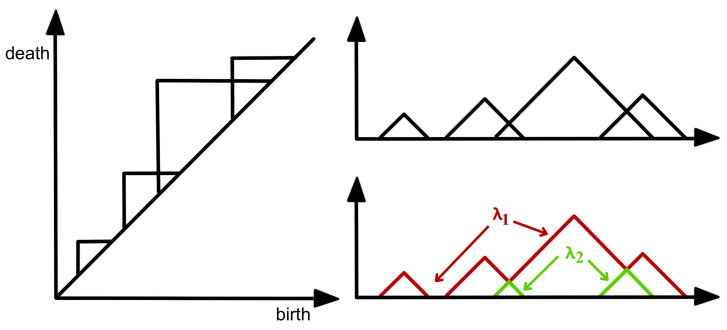
Construction of a complex persistence landscape (PL) from a persistence diagram (PD).

**Figure 14 entropy-25-01509-f014:**
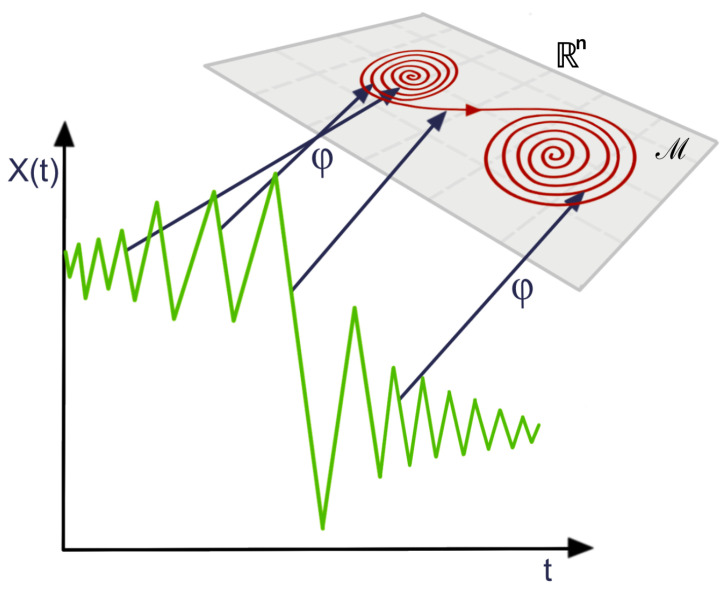
Illustration of the reconstructed (using the embedding map ϕ) attractor (red curve in the manifold $M\subset R^{n}$) from the time series observations.

**Figure 15 entropy-25-01509-f015:**
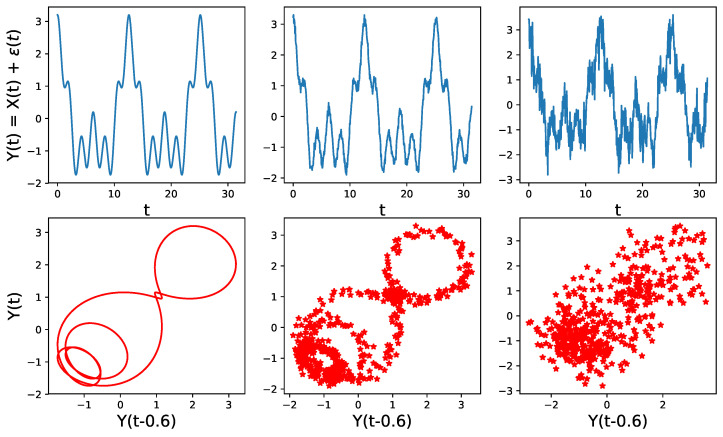
Point cloud embedding of a univariate time series using the sliding window method for different noise levels. Original time series (**TOP**) for various standard deviations of the noise ϵ (from left to right σ=0, σ=0.1, σ=0.5), with corresponding time-delay embedding (**BOTTOM**). The dependence structure in this time-delay embedding cannot be visually observed even in the presence of a moderate level of noise.

**Figure 16 entropy-25-01509-f016:**
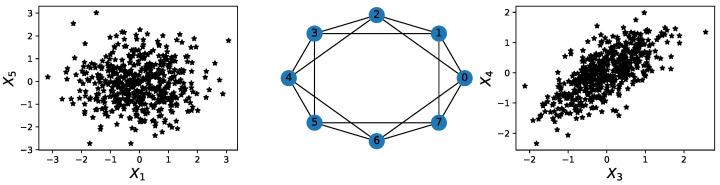
Illustration of a hidden cyclic pattern in the dependence structure. Left subplot: scatter plot between time series components $X_{5}$ and $X_{1}$. Right subplot: scatter plot between time series components $X_{4}$ and $X_{3}$. Middle subplot: the cyclic latent dependence network of the entire multivariate time series.

**Figure 17 entropy-25-01509-f017:**
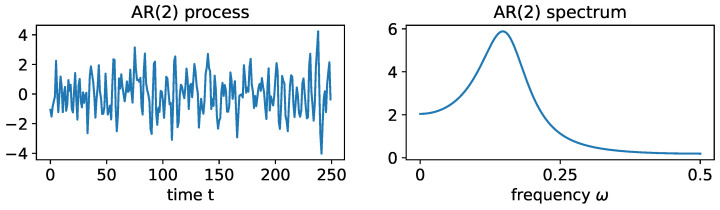
**LEFT**: realization from an AR(2) process with $\phi_{1}=\frac{2}{1.414}\cos\left(\pi \frac{154}{500}\right)$ and $\phi_{2}=-1/1.414^{2}$. **RIGHT**: true spectrum of this AR(2) process.

**Figure 18 entropy-25-01509-f018:**
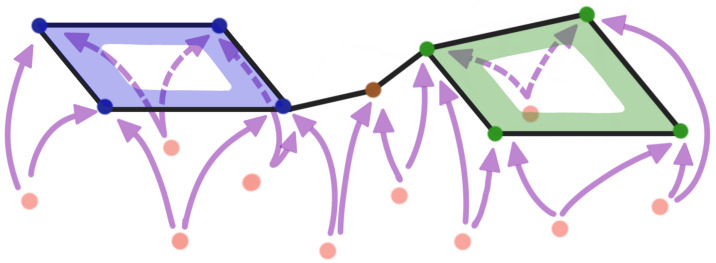
Example 1: Multivariate time series dependence network with two cycles pattern as defined in Equation (9).

**Figure 19 entropy-25-01509-f019:**
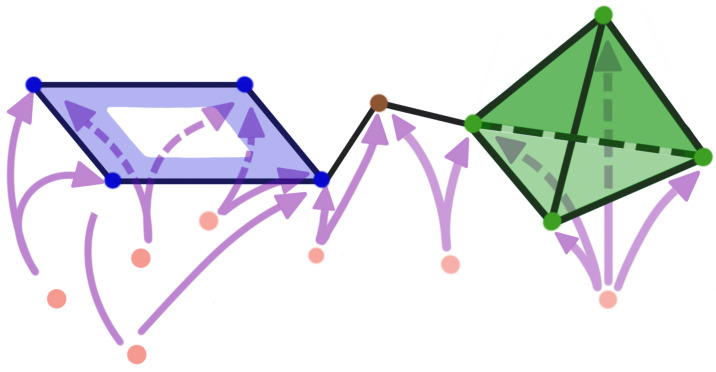
Example 2: Multivariate time series dependence network with a cycle and a 4-clique pattern, as defined in Equation (11).

**Figure 20 entropy-25-01509-f020:**
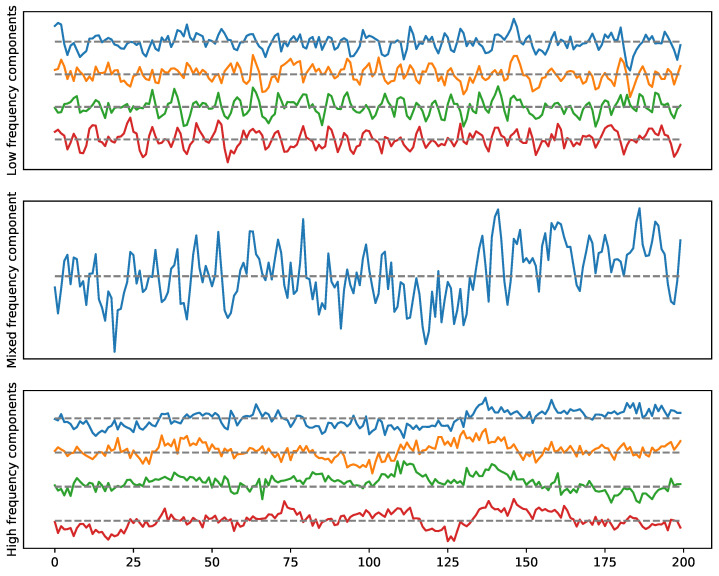
Example 1: A multivariate time series with a two-loop dependence pattern. High frequency in the top (cycle), low frequency in the bottom (cycle).

**Figure 21 entropy-25-01509-f021:**
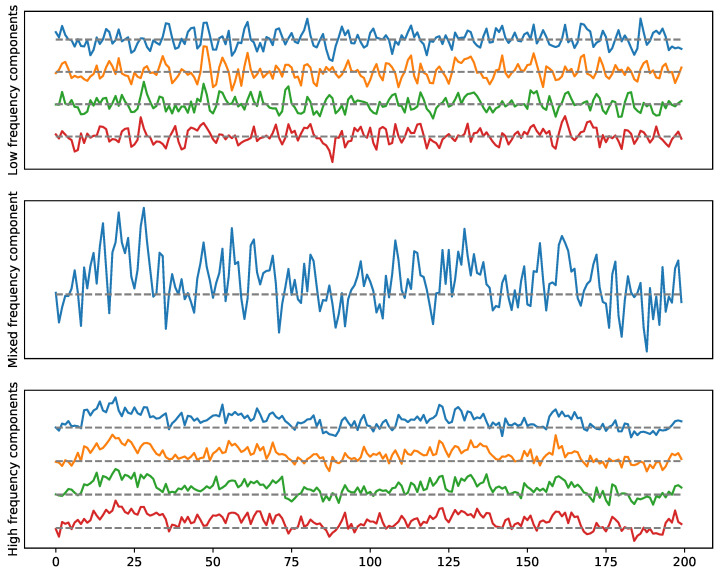
Example 2: A multivariate time series with one loop and a 4-clique dependence pattern. High-frequency in the top (cycle), low-frequency in the bottom (4-clique).

**Figure 22 entropy-25-01509-f022:**
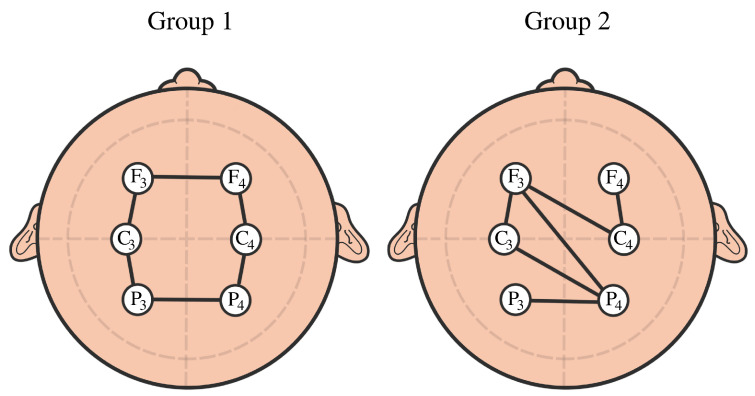
Example of cyclic brain connectivity vs. random connectivity.

**Figure 23 entropy-25-01509-f023:**
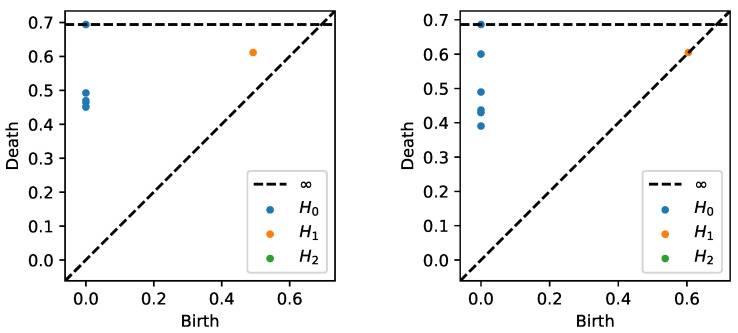
Persistence diagram based on the previous example displaying cyclic or group 1 (**LEFT**) and random brain connectivity or group 2 (**RIGHT**).

**Figure 24 entropy-25-01509-f024:**
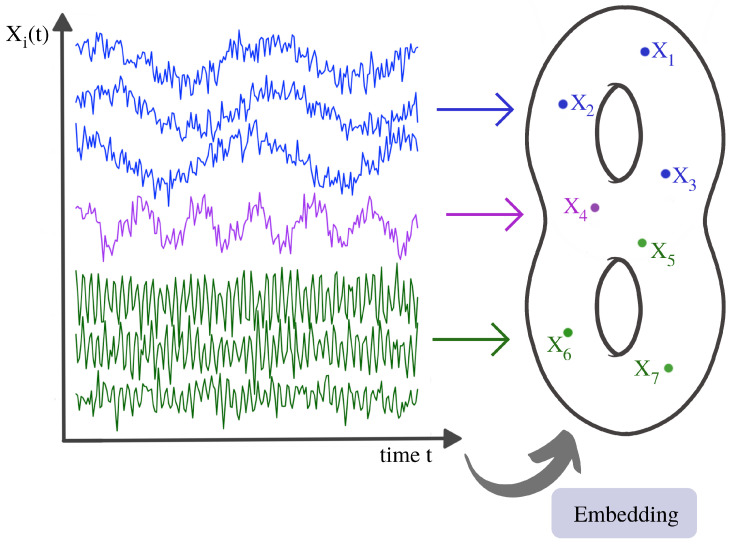
Illustration of a time series dependence embedding in some abstract dependence space.

**Figure 25 entropy-25-01509-f025:**
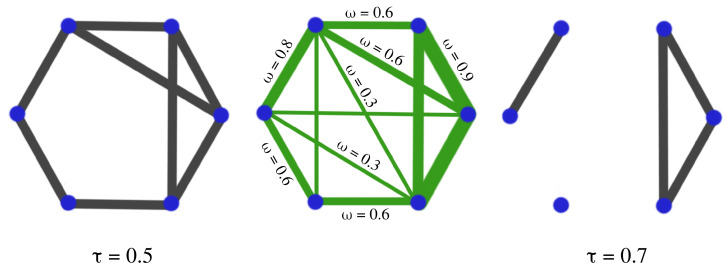
The thresholding step consists of comparing the weights of the original network (**CENTER**) with a given threshold τ. If the weight of the edge is larger than the threshold, one edge is created in the new network, if the weight is smaller, no edge is added. If the selected threshold is low (τ=0.5), the resulting network is dense (**LEFT**); if the threshold is too high (τ=0.7), the resulting network is sparse (**RIGHT**). The problem is now as follows: How do we select the threshold τ so we balance the loss of information with sparsity?

**Figure 26 entropy-25-01509-f026:**
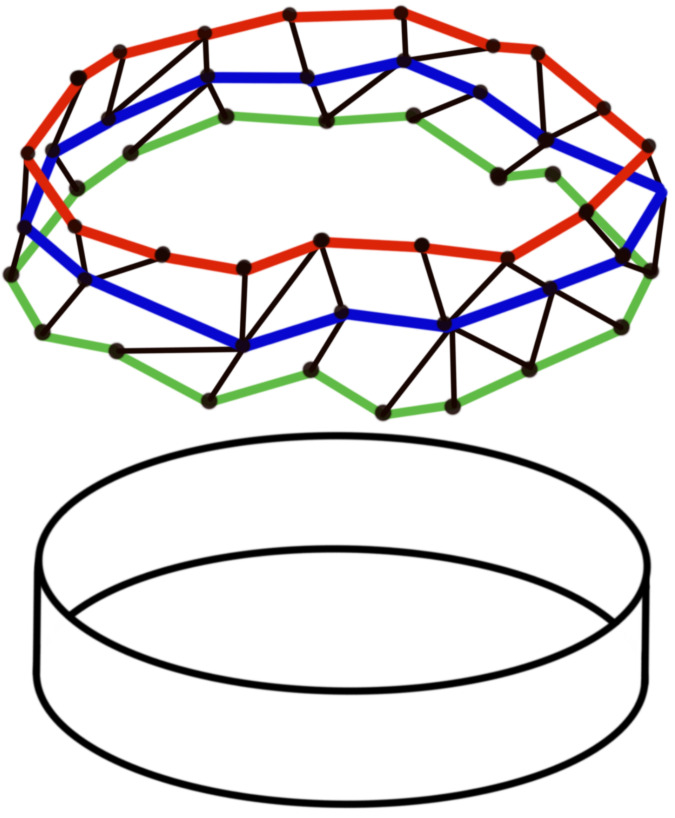
Network with a cyclic feature. Original network (**TOP**), topological structure being represented (**BOTTOM**).

**Figure 27 entropy-25-01509-f027:**
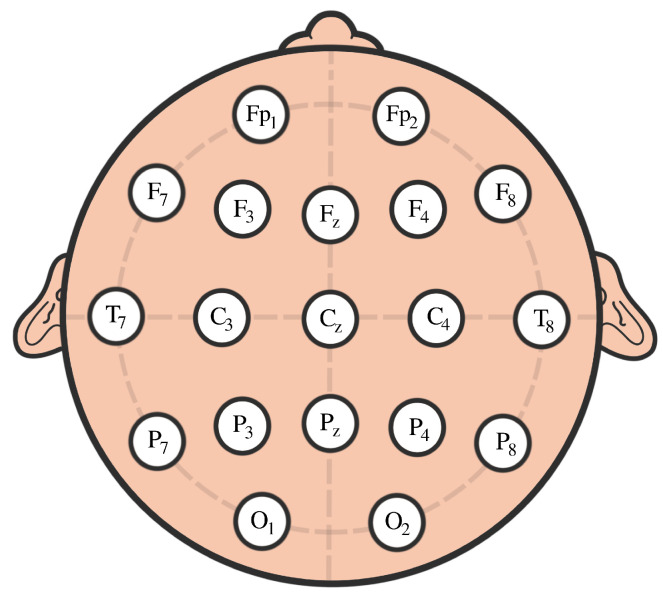
Scalp EEG with 10–20 standard layouts.

**Figure 28 entropy-25-01509-f028:**
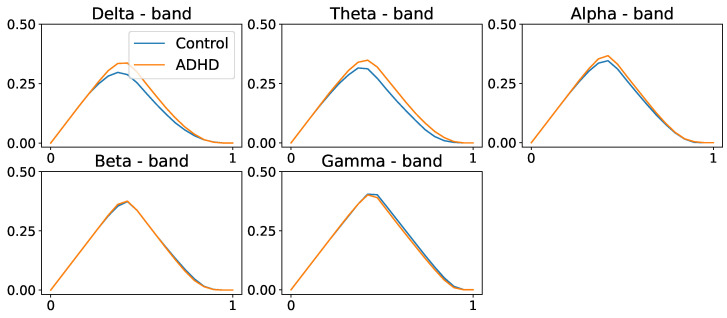
Population average persistence landscapes for the 0-dimensional homology group for ADHD (orange) and control (blue) groups, at various frequency bands. High-frequency bands do not seem to display any differences between the two groups. These plots suggest that both groups have a similar structure at the connected components level.

**Figure 29 entropy-25-01509-f029:**
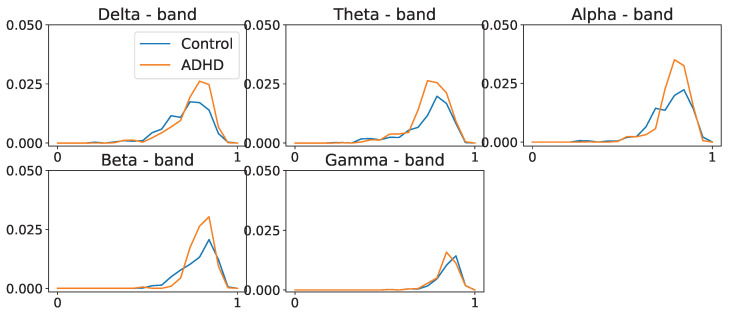
Population average persistence landscapes for the 1-dimensional homology group for ADHD (orange) and healthy control (blue) groups, at various frequency bands. Middle-frequency bands seem to display differences between the two groups. This suggests that the ADHD group seems to have more cycles/holes in their dependence network.

**Figure 30 entropy-25-01509-f030:**
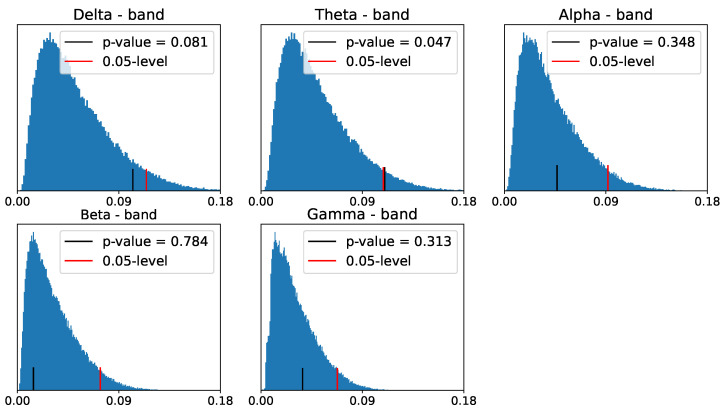
Reference distribution for testing for group-level differences between ADHD and healthy control persistence diagrams, based on B = 100,000 permutations. Zero-dimensional homology group.

**Figure 31 entropy-25-01509-f031:**
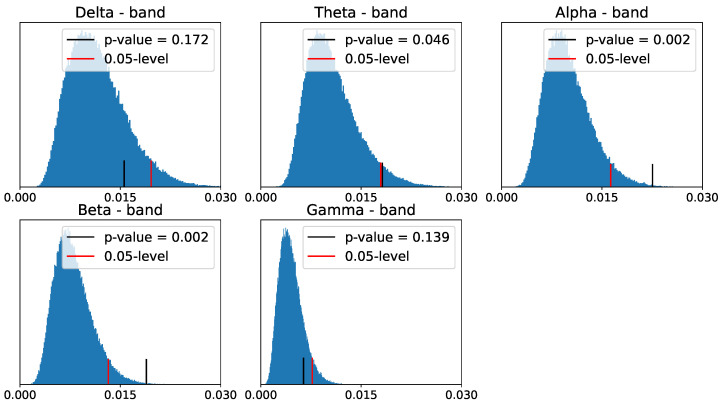
Reference distribution for testing for group-level differences between ADHD and healthy control persistence diagrams, based on B = 100,000 permutations. One-dimensional homology group.

**Table 1 entropy-25-01509-t001:** Advantages and Disadvantages of neuroimaging data analysis methods.

Method	Advantages	Disadvantages
CNN at the voxel level	Strong ability to capture spatial patterns. Can handle 2D or 3D datasets with high resolution. Automatically learns relevant features from raw data.	Requires large amounts of data for training. Model selection is difficult. Does not lend well to statistical analysis. Considers spatial neighbor only.
GNN Based on FC	Models brain networks as graphs. Leverages message passing across neighbors to learn the suitable representation. High discriminative power. Can detect topological differences between networks.	Requires large amounts of data for training. Model selection is difficult. Does not lend well to statistical analysis. Can be sensitive to node relabeling.
Morse Filtration	Provides topological insights on the arrangement of the local extrema. Can handle 1D signals (functions), 2D and 3D signals (images).	Assumes smooth functions. Cannot handle more than one time series at a time.
Time-Delay Embedding	Enables the topological analysis of time series data as a cloud point. Strong foundations from dynamical systems.	Hyperparameter selection (time lag and embedding dimension). Can be sensitive to noise.
Vietoris–Rips Filtration	Considers all threshold values. Quantifies topological patterns precisely at various dimensions. Robust to node relabeling. Can lend well to statistical analysis.	Can be computationally expensive for large datasets. Analyzes the data globally and may lack direct interpretability. Lack of directionality.

## Data Availability

The data were made available by Ali Motie Nasrabadi from Shahed University, by authors Armin Allahverdy, Mehdi Samavati, and Mohammad Reza Mohammadi, on 12 March 2022. Researchers interested in accessing the data can find it at https://ieee-dataport.org/open-access/eeg-data-adhd-control-children (accessed on 12 March 2022).

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
