# Peer review of "Topological Data Analysis for Multivariate Time Series Data"

_entropy, 2023, doi:10.3390/e25111509_

Round 1

Reviewer 1 Report

This paper employs topological data analysis to analyze multivariate time series data, emphasizing its application to brain signals and connectivity networks.

1. The authors assert that graph theoretical modeling, while providing valuable insights into the topological properties of brain networks, has inherent limitations, such as the inability to capture all topological information, the need for thresholding, and potential information loss. They propose that applying Topological Data Analysis (TDA) to brain connectivity can offer complementary insights since TDA considers all potential threshold values. However, it's worth noting that the field of brain connectivity has seen significant advancements, particularly with the advent of Graph Neural Networks (GNNs). For instance, some GNNs with attention mechanisms have demonstrated the capability to handle topological information in brain networks, even addressing the presence of holes and voids. Moreover, adaptive learning approaches have been proposed to address the thresholding issue, ensuring that important information is not lost. The limitations mentioned by the authors regarding previous studies in this field are not convincing. Therefore, it is essential to reconsider the motivation and contribution of this study in light of these existing capabilities and potential alternatives.

2. I strongly recommend including a comparison table with other relevant articles. This will significantly improve the comprehensiveness of the analysis of the proposed approach.

3. Please incorporate more up-to-date references into the study.

4. There is inconsistency in the font sizes used in the figures, with some appearing excessively large while others are too small.

The manuscript's language quality is acceptable. Overall, this paper is clear and well-structured.

Author Response

We would like to thank the paper reviewers for their time, valuable suggestions, and constructive remarks. These inputs have notably improved the paper, and we have diligently addressed each point, incorporating the necessary revisions into the manuscript.

Reviewer 2 Report

The article aims to provide an instructive overview of topological data analysis (TDA) in the context of time series data analysis using multivariate spectral analysis, offering the benefit of integrating the capabilities of TDA with spectral analysis. This integration allows us to describe both similarities and distinctions in the patterns of brain connectivity networks between various groups with a focus on frequency-specific neural oscillations.

The article is well written and adequately illustrates how TDA can be applied in the area of connectivity network analysis. I think it may be of great interest to those who work in this field since it solves some of the limitations of traditional methods.

I recommend its publication in its present form.

Author Response

We would like to extend our appreciation to the reviewer for their valuable support in endorsing the publication of our article in its current form.